# Effects of IMAZALIL on the Storage Stability and Quality of ‘Sefri Ouled Abdellah’ and ‘Kingdom’ Pomegranate Varieties

**DOI:** 10.3390/foods14030337

**Published:** 2025-01-21

**Authors:** Chaimae El-Rhouttais, Zahra El Kettabi, Salah Laaraj, Abdelaziz Ed-Dra, Samir Fakhour, Ammadi Abdelillah, Kaoutar Elfazazi, Souad Salmaoui

**Affiliations:** 1Agro-Food Technology and Quality Laboratory, Regional Center of Agricultural Research of Tadla, National Institute of Agricultural Research, Avenue Ennasr, BP 415 Rabat Principale, Rabat 10090, Morocco; chaimae.elrhouttais@gmail.com (C.E.-R.); elkettabi.zahra@gmail.com (Z.E.K.); salah.laaraj@usms.ma (S.L.); ammadiabdelillah@gmail.com (A.A.); 2Environmental, Ecological and Agro-Industrial Engineering Laboratory, LGEEAI, Faculty of Science and Technology (FST), Sultane Moulay Slimane University (USMS), Beni Mellal 23000, Morocco; souadsalmaoui@yahoo.fr; 3Laboratory of Engineering and Applied Technologies, Higher School of Technology, M’ghila Campus, Sultan Moulay Slimane University (USMS), Beni Mellal 23000, Morocco; abdelaziz_iaa@yahoo.fr; 4Plant Protection Laboratory, Regional Center of Agricultural Research of Tadla, National Institute of Agricultural Research, Avenue Ennasr, BP 415 Rabat Principale, Rabat 10090, Morocco; sfakhour@gmail.com

**Keywords:** pomegranate (*Punica granatum* L.), imazalil, cold storage, nutritional quality, bioactive compounds

## Abstract

Employing post-harvest treatments to maintain pomegranate fruit quality during storage is a prevalent practice within the food industry. IMAZALIL (IMZ), a fungicide, has demonstrated efficacy in reducing both the incidence of chilling injury symptoms and the presence of pathogenic fungi. This study aims to assess the impact of IMZ treatment on the technological quality (weight loss, color attributes (C* and h°), pH, titratable acidity, and total soluble solids), nutritional properties (total sugars content), and functional properties (total phenolic compounds (TPC) and total anthocyanin content (TAC)) in pomegranate fruits of the ‘Sefri Ouled Abdellah’ and ‘Kingdom’ cultivars. These fruits were collected in the Beni Mellal region and immediately stored at 4 °C for 120 days. Untreated pomegranates exhibited significant degradation in overall quality when stored in cold conditions. The fruits treated with IMZ are characterized by a major loss in weight (3.41% to 20.11%) compared to the control fruits (1.62% to 13.19%). This was accompanied by more pronounced color degradation in the IMZ-treated fruits relative to the control. This study substantiates the effectiveness of IMZ treatment in prolonging the post-harvest quality of pomegranates during cold storage, demonstrating superior efficacy in delaying losses in bioactive compounds by 39.44% and enhancing nutritional properties by 18.84%. This finding initiates the exploration of optimal IMZ concentrations and the best treatments to maintain the overall quality of pomegranate fruits.

## 1. Introduction

The pomegranate (*Punica granatum* L.) is one of the most ancient and revered edible fruits, cultivated in various geographical regions, including Morocco, and typically consumed as juice or in its fresh form. Interest in pomegranate juice and its derivative food products has significantly increased in recent years, driven by a growing body of literature extolling its potential health benefits [1,2]. This surge has propelled pomegranate to the forefront of the functional juice industry, prized for its abundance of phenolic compounds such as flavonoids (including anthocyanins and flavonols), condensed tannins (proanthocyanidins), and hydrolysable tannins (ellagitannins and gallotannins) [3]. Moreover, pomegranate juice boasts a wealth of sugars, organic acids, vitamins, polysaccharides, and essential minerals [2,4].

Fully ripe harvested pomegranate fruits are highly susceptible to various physiological and enzymatic disorders, presenting significant challenges during storage. Common issues include pronounced weight loss, husk browning, and surface pitting extending to the arils, ultimately resulting in a decline in overall quality during post-harvest storage [5]. The recommended storage temperature to preserve overall quality is below 5 °C [6]. However, maintaining such low temperatures also presents a significant challenge, as it can worsen issues related to dehydration and chilling injury, resulting in the deterioration of both external appearance and internal fruit quality [7]. To mitigate the adverse effects of cold storage on pomegranate fruit quality, various strategies have been explored. These include biological treatments such as spermidine [8], chemical treatments like fludioxonil [9], hormone treatments such as benzyladenine [10], controlled and modified atmosphere storage, intermittent warming [11], as well as the use of shrink films wrapping and coatings [12]. Among these, Castillo et al. [13] reported that the combined application of wax and chemical treatment such as Imazalil significantly enhances fruit quality preservation compared to wax alone. This finding underscores the critical role of Imazalil in maintaining fruit quality during cold storage, further emphasizing its potential as an effective postharvest treatment.

Imazalil (IMZ) is a widely used fungicide that has garnered attention for its efficacy in preserving the quality of fresh fruits during storage. It is classified as having moderate toxicity; however, its safety for human health is primarily ensured when applied within regulatory limits set by authorities such as the Codex Alimentarius, the European Food Safety Authority (EFSA), and the U.S. Food and Drug Administration (FDA). These regulations establish Maximum Residue Limits (MRLs) for IMZ, ensuring that its concentration on treated fruits remains below levels considered safe for consumption (Castillo et al., 2014 [13]). According to Commission Regulation (EU) 2022/1290, the safety of pomegranates for human health is ensured when the MRL is set at or below 0.01 mg/kg [14]. The application of IMZ has been extensively studied across various fruit types, including citrus, apples, pears, and pomegranates [10,15,16]. The IMZ treatment has shown promising results in inhibiting fungal growth (such as *Penicillium digitatum*) and extending the shelf life of fruits by reducing postharvest decay [17,18]. Furthermore, research indicates that IMZ treatments can maintain the visual appearance, texture, and flavor of various fruits, thereby enhancing consumer appeal and marketability. However, to our knowledge, little is known about the effect of IMZ on preserving the morphological, functional, and nutritional properties of pomegranate fruits during cold storage. In this regard, this study aims to evidence the advantages of using IMZ during postharvest storage of pomegranate fruits under cold temperatures, by exploring its effect on the preservation of morphological (weight loss and color attributes), physicochemical (pH, titratable acidity, and total soluble solids), nutritional (total sugars content), and functional (total phenolic compounds and total anthocyanins content) properties during postharvest storage of pomegranates under cold conditions.

## 2. Materials and Methods

### 2.1. Plant Material

Two cultivars of fresh pomegranates ‘Kingdom’, and ‘Sefri Oulad Abdellah’ were harvested in November of 2021 in the Beni Mellal region of central Morocco (23°50′05″ E; 6°48′98″ N) and immediately transported to the laboratory. A total of 160 fully ripe pomegranate fruits were randomly selected and divided into two batches of 80 fruits each.

The fungicide used was a commercially available formulation of IMZ (FUNGAFLOR 500 EC) with a concentration of 50% active ingredient.

### 2.2. Experimental Design

The received fruits were sorted, washed with cold tap water, and prepared for the treatment process. Postharvest treatment involved immersing the fruits in 0 (control) and 2000 µL/L IMZ solution for 5 s at 25° C. Subsequently, the fruits were removed from the IMZ solution, air-dried at room temperature, and then stored at 4 ± 1 °C (85–90% RH) for 120 days.

Throughout the stored period at 4 °C, the pomegranate fruits were analyzed fortnightly on eight specific dates: 15, 30, 45, 60, 75, 90, 105, and 120 days, to evaluate the morphological (weight loss and color attributes), physicochemical (pH, titratable acidity, and total soluble solids), nutritional (total sugars content), and functional (total phenolic compounds and total anthocyanins) properties.

### 2.3. Morphological Properties

#### 2.3.1. Weight Loss Percentage

The weight loss was assessed by measuring the whole pomegranate at the end of each storage duration [19]. The formula used to calculate weight loss is represented as a percentage of weight loss relative to the initial weight before storage.WL = (Initial weight − Secondary weight)/(Initial weight × 100)

#### 2.3.2. Color Attributes

The color of pomegranate peels was determined in CIELAB coordinates (L*, a*, b*) using a Minolta Chroma Meter CR-400 (Minolta Corp., Osaka, Japan) [20,21]. Measurements were performed in a cold room to avoid temperature fluctuations (4 °C). Peel color measurements were used to monitor changes in the external color of the fruit. These measurements were taken at three designated surface areas located around the middle of the fruit. The color parameters of chroma (C) and hue angle (h°) were calculated using the following equations:C* = (a ^(^*^2)^ + b ^(^*^2)^)^(1/2)^h° = tan^(−1)^ (b/a)

### 2.4. Physicochemical Properties

Nine fruits per cultivar were sampled for physicochemical analysis (pH, titratable acidity, and total soluble solids). Juice was extracted using a juice extractor (Mellerware, Cape Town, South Africa). The pH of the juice was determined at room temperature using a pH meter (Thermo Orien 3 star, Waltham, MA, USA). Titratable acidity (TA) was determined by titrating 10 mL of juice with 0.1 N sodium hydroxide (NaOH) to a pH of 8.1, and results were expressed as g citric acid equivalent/100 mL. Total soluble solids (TSS) were measured using a digital refractometer (Metteler-Toledo GmbH, Greifensee, Switzerland).

### 2.5. Nutritional Properties

#### Total Sugars Content

The quantification of total sugars was conducted using the phenol sulfuric acid method as outlined by Dubois et al. [22]. Briefly, 1 mL of juice diluted at a ratio of 1:500 with distillate water was mixed with 5 mL of concentrated sulphuric acid (5%) and 1 mL of phenol (5%). The entire mixture was vortexed before being left to settle at room temperature for 40 min. The measurement of absorbance was performed using a UV spectrophotometer (serial N° A 10834232128CS, Suzhou Instruments Manufacturing, Suzhou, China) at a wavelength of 488 nm. The total sugar concentrations were determined using the glucose standard curve method at a wavelength of 485 nm and the results are expressed as g·100 mL^−1^ juice.

### 2.6. Functional Properties

#### 2.6.1. Total Phenolic Compounds (TPC)

The quantification of total phenolic content was conducted using the Folin–Ciocalteu method, as described by Singleton and Rossi [23]. Briefly, 300 μL of juice was diluted with a mixture of methanol and water (6:4) at a ratio of 1:20. This diluted juice was then combined with 1.2 mL of 7.5% sodium carbonate and 1.5 mL of Folin–Ciocalteu reagent, which was diluted at a ratio of 1:10. Following a 90-min reaction at ambient temperature, the absorbance was measured using a UV-Visible spectrophotometer (Schimadzu-UV-2401PC; serial N° A 10834232128CS, Suzhou Instruments Manufacturing, Suzhou, China) at a wavelength of 760 nm. The results were expressed as milligrams of gallic acid equivalent per 100 milliliters of juice.

#### 2.6.2. Total Anthocyanin Content (TAC)

The Total Anthocyanin Content (TAC) was measured following the protocol outlined by Ozgen et al. [24], employing two buffer systems: sodium acetate buffer with a pH of 4.5 (0.4 M) and potassium chloride buffer with a pH of 1.0 (25 mM). In summary, a total of 0.4 mL of pomegranate juice was mixed with 3.6 mL of buffers that corresponded to it. The absorbance of the mixture was then measured at wavelengths of 510 and 700 nm. Water served as a blank. The absorbance (A) was represented by the following equation:A = (A (520 nm) − A_(700 nm))_〖pH〗1.0_ − (A_(520 nm) − A_(700 nm))_〖pH〗4.5_

The TAC of the juice was expressed as milligrams of cyanidin-3-glucoside per liter of pomegranate juice, and was calculated using the following equation:TAC = (A × MW × 100) × 1/MA
using the following abbreviations: A for absorbance, MW for molecular weight (449.2 g moL^−1^), and MA for molar absorptivity of cyanidin-3-glucoside (26.900).

### 2.7. Statistical Analysis

All measurements were performed in triplicate. All the studied variables are expressed as mean ± standard error (S.E.). An analysis of variance (MANOVA) was performed using SPSS software version 26 (IBM Software, Inc., Armonk, NY, USA).

## 3. Results and Discussion

The analyses of variance showed a highly significant (*p* < 0.001) effect of the cultivar, treatment, duration of storage, and their interaction on all the morphological, physicochemical, nutritional, and functional properties (Table 1).

### 3.1. Morphological Properties

#### 3.1.1. Weight Loss Percentage

The weight loss of the fruits underwent significant change (*p* < 0.001) with both storage duration and treatment applied. The data presented in Figure 1 clearly indicate that the percentage weight loss of pomegranate fruits stored at 4 °C and 85–90% (RH) significantly increased (*p* < 0.001) with prolonged storage time. Furthermore, postharvest treatment with IMZ significantly heightened the weight loss incidence of the stored fruits compared to untreated fruits. The highest percentage of weight losses (22.37% and 17.83%) were observed in the IMZ-treated fruits of ‘Kingdom’ and ‘Sefri Oulad Abdellah’ pomegranate cultivars, respectively, at the end of the storage period, while the lowest percentage weight losses (12.72% and 13.57%) were recorded in the untreated fruit of both cultivars.

The lipophilic nature of IMZ treatment typically serves as a permeable barrier to oxygen, reducing respiration, water loss, and oxidation reaction rates. However, the use of IMZ in the postharvest treatment of pomegranates appears to be associated with increased water loss from the fruits. This observation could be attributed to various factors, including physiological changes in the fruit, interactions with the cuticle, or effects related to the formulation and concentration of IMZ. Additionally, pomegranate is highly susceptible to weight loss due to the high porosity of the fruit skin, allowing free movement of water vapor [25], as well as loss through respiration and transpiration processes [6].

These results are consistent with findings from studies of other fruit cultivars. For instance, Aborisade and Romero et al. [26,27] noted that the percentage weight loss of orange fruits treated with IMZ increased gradually and significantly with prolonged storage time. Similar patterns of change were reported for grapefruit [28], tangerines [29], and sweet cherries [30]. Contrary weight loss was significantly reduced in lemons coated with wax + IMZ [13], attributed to the wax’s function of sealing natural openings on the surface of the fruit, inhibiting the diffusion of water vapor and resulting in slower moisture loss [31].

#### 3.1.2. Color Attributes

Table 2 shows the C* (color intensity) and h* (Hue Angle) of both treated and untreated pomegranate fruit peels during the storage period. The results showed significant changes (*p* < 0.001) in the fruit skin color intensity of both ‘Sefri Ouled Abdellah’ and ‘Kingdom’ cultivars during the storage period. The application of the IMZ treatment notably decreased significantly (*p* < 0.001) the color intensity of the fruit compared to the untreated ones. By the end of the storage period, the mean value of C* was significantly higher (*p* < 0.001) in the untreated cultivar ‘Sefri Ouled Abdellah’ than in the IMZ-treated fruits, with final values of 27.73 ± 1.77 and 23.15 ± 2.00, respectively. Similarly, in the ‘Kingdom’ cultivar, the C* value at the end of the storage period was significantly lower (*p* < 0.001) in the IMZ-treated fruits compared to the untreated fruits, with final values of 20.24 ± 1.77 and 23.22 ± 2.46, respectively.

The h* value of the fruits also showed a significant difference (*p* < 0.001) in the IMZ-treated cultivars ‘Sefri Ouled Abdellah’ and ‘Kingdom’ during storage. In ‘Sefri Ouled Abdellah’, the IMZ treatment effectively maintained the stability of the h* value, with minimal deviation observed during the storage period. However, applying the treatment to the cultivar ‘Kingdom’ resulted in a significant change (*p* < 0.001) in the h* value (21.52 ± 4.51 to 1.29 ± 1.21), following a similar trend observed for the C* values during the storage period.

This difference can be attributed to the effect of the IMZ treatment, which is widely used as a postharvest fungicide known for its ability to inhibit fungal growth and delay fruit senescence. By reducing microbial activity and oxidative stress, IMZ reduces biochemical changes, such as the degradation of pigments or changes in the fruit’s surface structure, that influence the C*and h° parameters. Consequently, the treatment preserves the fruit’s overall color stability, resulting in lower C* and h° values than untreated fruits.

These results are consistent with the findings of previous studies, reporting that the skin hue angle varies during cold storage [32]. In a similar study, the treatment of lemons ‘Kuetdiken’ with IMZ showed a slight change in fruit color during 12 weeks of storage [33]. However, in another study, Risse et al. [34] reported that the IMZ treatment did not affect the color development of cucumbers during storage.

### 3.2. Physicochemical Properties

During storage, the pH value increased significantly (*p* < 0.0001) in both the IMZ-treated and untreated pomegranates ‘Sefri Ouled Abdellah’ and ‘Kingdom’. A significant increase in the pH value (*p* < 0.0001) was observed in the control fruit ‘Kingdom’ after 15 days of storage (Figure 2). However, the pH values of the treated fruit generally remained stable until the 90th day of storage. In ‘Sefri Ouled Abdellah’, the IMZ treatment maintained a lower pH value (4.36 ± 0.12) than in the control group (4.46 ± 0.18) at the end of the storage period. These changes in fruit pH values during storage could be an indication of the content and dynamics of fruit organic acids [35]. Additionally, as storage progresses, metabolic reactions and microbial activities may further influence pH levels. Therefore, this effect can be attributed to the antifungal properties of IMZ, which inhibit microbial activity responsible for the degradation of organic acids in fruits. The degradation of organic acids is an important factor contributing to pH increases during storage. By inhibiting microbial growth, IMZ effectively reduces the degradation of these acids, thereby maintaining a more stable pH and preserving fruit quality during storage. Several studies have confirmed that the pH value in pomegranate fruit increases during storage [36,37].

Juice TSS was significantly (*p* < 0.0001) affected by the IMZ treatment and the storage time (Figure 3). For the ‘Kingdom’ control, the mean TSS value decreased significantly throughout cold storage (14.83 ± 0.56 to 13.39 ± 0.87%) in the first month of storage, followed by the stability of the TSS for about three months, which decreased again in the last week of the storage period, reaching 13.06 ± 0.58%. However, the ‘Kingdom’ pomegranate treated with IMZ showed a similar trend, with the TSS value remaining stable for about two months and then starting to decrease to reach a low TSS value (12.27%) compared to the control fruit (13.05%). On the contrary, the mean TSS value increased significantly (*p* < 0.0001) with increasing storage time in both the IMZ-treated and untreated ‘Sefri Ouled Abdellah’ pomegranates. The TSS value at the end of the storage period was significantly higher (*p* < 0.001) in the control fruit (15.33%) than in the fruit treated with IMZ (13.78%). This effect is due to the antifungal properties of IMZ, which inhibit microbial activity and slow down metabolic processes, such as the hydrolysis of polysaccharides into simple sugars. The IMZ treatment effectively maintains lower TSS levels by delaying senescence and sugar accumulation, thereby preserving the fruit’s quality during storage. The results thus obtained for ‘Kingdom’ are in line with previous studies [38,39,40] which reported a decrease in TSS during storage. Fawole et al. [6] explained that this decrease could be due to the degradation of sugars with a prolonged storage period. On the contrary, the findings for ‘Sefri Ouled Abdellah’ are in agreement with those of Alighourchi et al., Arendse et al. and Elfazazi et al. [36,41,42] who reported an increase in TSS with increasing storage time. The possible reason for the observed increase in TSS content could be a consequence of moisture loss, leading to a concentration of sugars inside the fruit [36]. Castillo et al. [13] reported a significant decrease in the TSS of lemons treated with wax-IMZ. On the other hand, the TSS of pears treated with IMZ increased with increasing storage time [43].

The mean TA of pomegranate fruit decreased significantly with increasing storage time in both the IMZ-treated and untreated pomegranates ‘Sefri Ouled Abdellah’ and ‘Kingdom’ (Figure 4). In ‘Kingdom’, the TA value at the end of the storage period was significantly (*p* < 0.001) lower in the IMZ-treated fruit (0.38% citric acid) than in the control fruit (0.44% citric acid). In contrast, the TA value of ‘Sefri Ouled Abdellah’ was not significantly affected by the IMZ treatment, indicating that the effect of IMZ on TA may depend on the specific metabolic characteristics of each cultivar. Decreases in TA levels during storage have previously been reported by Arendse et al. and Alighourchi et al. [36,41]. These results are in line with previous studies [6,40] which reported a significant decrease in TA levels for pomegranate fruits during storage. On the contrary, several studies reported an increase in TA with increasing storage time [6]. In addition, a similar study found a significant decrease in TA levels for lemon [13] and pear fruit [43] treated with IMZ.

### 3.3. Nutritional Properties

#### Total Sugars Content

The total sugar content (TSC) showed significant changes (*p* < 0.0001) during storage in IMZ-treated and untreated pomegranates of both cultivars (Figure 5). During storage, TSC showed different behavior in ‘Sefri Ouled Abdellah’ and ‘Kingdom’. For instance, in ‘Sefri Ouled Abdellah’, the TSC value increased in the control fruit (79.04 ± 1.11 to 87.21 ± 3.93 mg/100 mL), while it decreased in the treated fruit during the first three months of storage and then increased again to reach a value (88.69 mg/100 mL) close to that of the control fruit (87.21 mg/100 mL). In addition, the TSC value at the end of the storage period showed an almost similar concentration in the IMZ-treated (81.11 mg/100 mL) and untreated (80.62 mg/100 mL) ‘Kingdom’ pomegranates. In ‘Sefri Ouled Abdellah’ treated with IMZ, there seemed to be an anomaly in the evolution of total sugars in the pomegranate fruit during storage. In our study, we discovered a deviation from the usual pattern of reducing total sugars. Instead, we found an initial decline followed by an increase after 3 months. This phenomenon could be due to complex enzymatic reactions or the conversion of starch into simple sugars at the beginning of the storage period. Stomatal closure is another factor that can contribute to the build-up of sugars in the fruit. Biale [44] reported that the increase in total sugars in the fruit was due to the conversion of starch to sugars by hydrolysis. Similar patterns of change have been reported for guava fruit [45], where TSC increased during storage. However, in soybean seeds, the TSC value decreased during storage [46].

### 3.4. Functional Properties

#### 3.4.1. Total Phenolic Compounds

The total phenolic compounds decreased significantly (*p* < 0.0001) in the IMZ-treated and untreated pomegranates of both cultivars. The IMZ-treated pomegranate ‘Sefri Ouled Abdellah’ showed a significant decrease (*p* < 0.0001) during the storage period (Figure 6). Meanwhile, the ‘Sefri Ouled Abdellah’ control exhibited a significant decrease in TPC during the first week of storage (290.11 ± 20.45 to 224.71 ± 17.26 mg GAE 100 mL^−1^), followed by the stability of TPC for about one month, which then experienced another decrease throughout the entire cold storage period, reaching 92.94 ± 25.91 mg GAE 100 mL^−1^. The TPC of the ‘Kingdom’ cultivar decreased during storage and reached a low concentration in the treated fruits (148.94 ± 18.96 mg GAE 100 mL^−1^) compared to the control fruits (174.68 ± 10.38 mg GAE 100 mL^−1^). These results can be explained by the fact that the fungicide IMZ can enter into chemical reactions with the polyphenols contained in pomegranate juice, leading to their degradation or the formation of less antioxidant compounds. Certain fungicides can also interfere with the enzymes responsible for the synthesis and stability of polyphenols, thus increasing the degradation of these compounds. In addition, fungicide treatment can alter the matrix of pomegranate, affecting its chemical composition and contributing to the degradation of polyphenols. The same behavior was previously reported by Arendse et al., Fialho et al. and Kaoutar et al. [40,47,48], who reported that TPC was significantly (*p* < 0.0001) affected by storage and decreased with storage duration. The subsequent reduction in the jus TPC could be related to the degradation of phenolic compounds [36,37]. However, the TPC increases significantly during storage time for ‘Wonderful’ pomegranate [40]. In the same experiment carried out by Hassan et al. [43], the pear fruit treated by IMZ showed a different response to the pomegranate fruits, with a significant increase in the TPC during the storage period.

#### 3.4.2. Total Anthocyanin Content (TAC)

The total anthocyanin content (TAC) was significantly (*p* < 0.0001) affected by both the IMZ treatment and the storage time. In the ‘Kingdom’ cultivar control, the mean TAC value decreased significantly (*p* < 0.0001) during the storage period (Figure 7). Meanwhile, the ‘Kingdom’ pomegranate treated with IMZ showed a stable TAC value during the first month of storage, which decreased throughout the cold storage period and reached a higher concentration (56.97 ± 3.09 mg·L^−1^) than in the control fruit (21.2 ± 5.5 mg·L^−1^) at the end of the storage period. The ‘Sefri Ouled Abdellah’ treated with IMZ showed a slight decrease in TAC during the 90th day of storage (86.21 ± 29.02 to 69.27 ± 7.05 mg·L^−1^). In contrast, the control fruit ‘Sefri Ouled Abdellah’ showed the highest TAC degradation (86.21 ± 29.02 to 40.08 ± 38.41 mg L^−1^). These results are consistence with Alighourchi et al., Arendse et al. and Fawole et al. [6,36,41], who reported that total anthocyanin concentration in pomegranate juice decreased significantly with increasing storage time. Alighourchi et al. [49] investigated the TAC content of selected pomegranate juices during storage at 4 °C for 120 days. Their results showed that the pomegranate juices stored at 4 °C exhibited less variation in anthocyanin content. Subsequently, in our study, the TAC content decreased significantly during storage, which can be attributed to a strong dependence on the variety and the treatment applied. The loss of TAC is probably due to oxidation as well as due to condensation of anthocyanin pigments with ascorbic acid [50].

## 4. Conclusions

Postharvest cold storage of pomegranate fruit can lead to various physiological and biochemical quality deteriorations that affect the color, flavor, and overall nutritional quality of the fruit. The storage conditions and the treatment applied are the key factors for maintaining quality during cold storage. Therefore, this study provides important information on the effect of IMZ on the preservation of physicochemical quality, peel color and bioactive compounds of two pomegranate cultivars during 120 days of cold storage at 4 °C.

Results of technological quality indicate that pomegranate fruits treated with a concentration of 2000 µL of IMZ exhibited significant degradation during cold storage, characterized by a major loss in weight loss and color degradation compared to the control fruits. These outcomes may be attributed to the effects associated with the applied treatment concentration. However, the treatment maintains the pH stability of the fruit during a significant period of storage. Regarding bioactive compounds, both control cultivars showed high losses in TPC and TAC during storage. IMZ was more effective in delaying the change, and loss for both cultivars and maintained stability in TAC during storage for ‘Sefri Ouled Abdellah’ cultivar.

Further investigations are required to elucidate the underlying mechanisms of this degradation in technological quality and to assess other concentrations of IMZ treatment on the overall quality of pomegranate fruits during storage. Additional studies are currently conducting to identify the best treatment tailored to specific cultivars. The aim is to preserve the overall quality of the fruit, ultimately reducing postharvest losses and prolonging the shelf life of pomegranate fruits.

## Figures and Tables

**Figure 1 foods-14-00337-f001:**
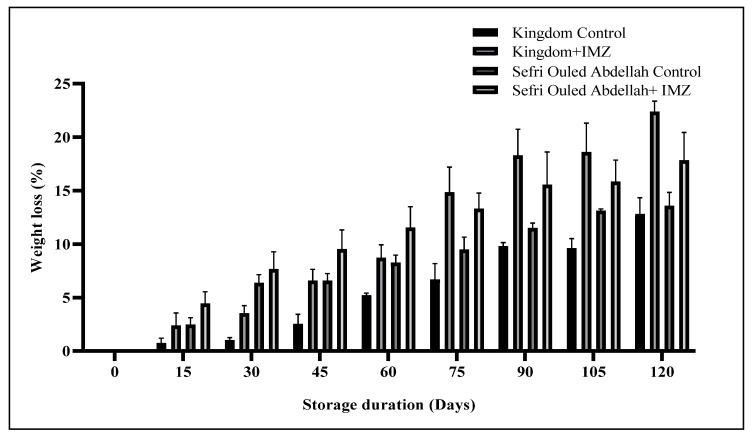
Changes in weight loss during the storage time of pomegranate cultivars.

**Figure 2 foods-14-00337-f002:**
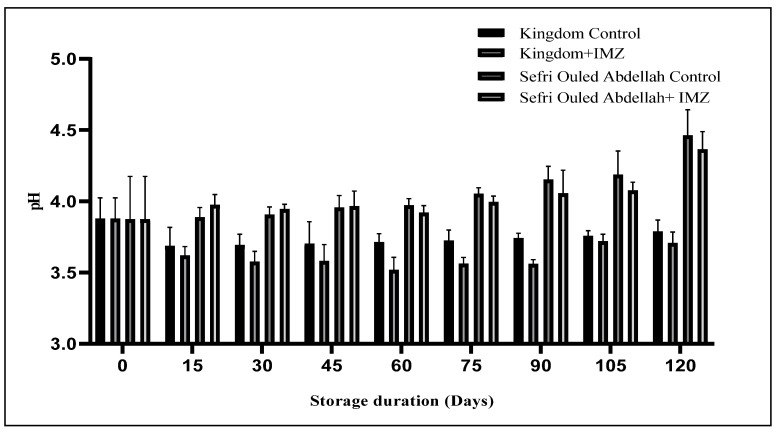
Changes in pH in pomegranate fruits after 120 days of storage at 4 °C.

**Figure 3 foods-14-00337-f003:**
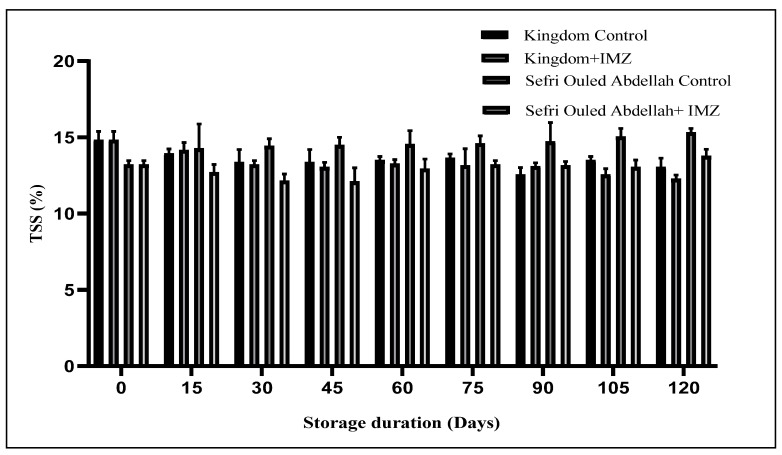
Changes in TSS in pomegranate fruits after 120 days of storage at 4 °C.

**Figure 4 foods-14-00337-f004:**
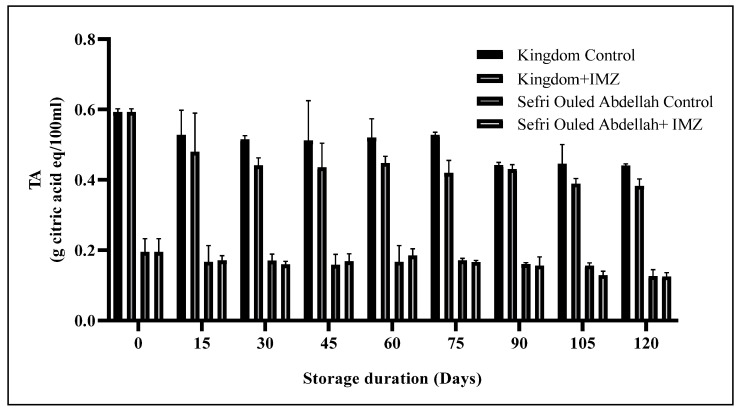
Changes in TA in pomegranate fruits after 120 days of storage at 4 °C.

**Figure 5 foods-14-00337-f005:**
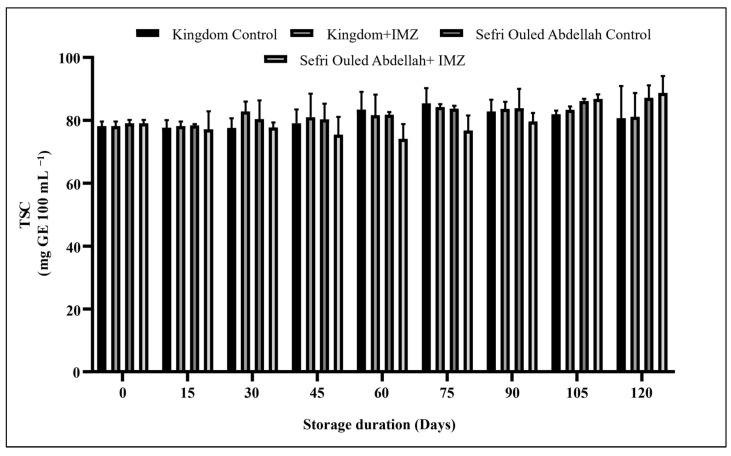
Changes in total sugar content (TSC) in pomegranate fruits over 120 days of storage at 4 °C.

**Figure 6 foods-14-00337-f006:**
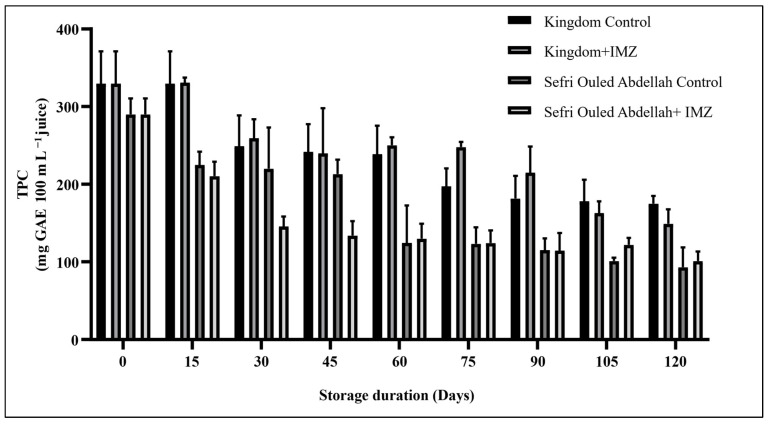
Changes in total phenolic compounds (TPCs) in pomegranate fruits over 120 days of storage at 4 °C.

**Figure 7 foods-14-00337-f007:**
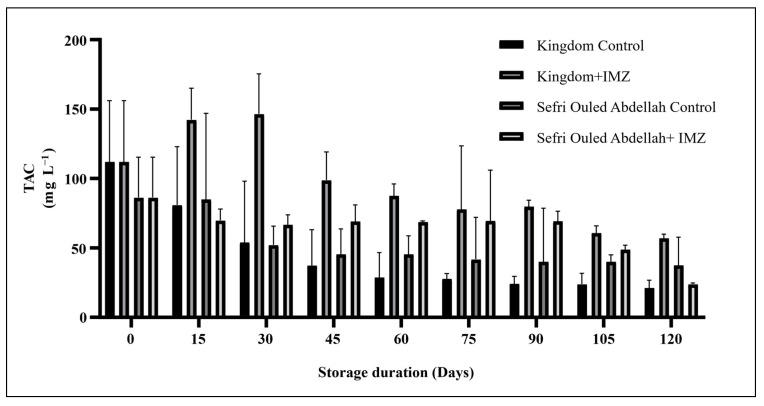
Changes in total anthocyanin content (TAC) in pomegranate fruits over 120 days storage at 4 °C.

**Table 1 foods-14-00337-t001:** Effect of cultivar, treatment and storage duration on the morphological, physicochemical, nutritional, and functional properties in pomegranate fruits (‘Kingdom’ and ‘Sefri Oulad Abdellah’) based on two-way MANOVA analysis.

Source	Dependent Variable	Type III Sum of Squares	Mean Square	F	Sig.	Partial Eta Squared
Corrected Model	Weight loss	11,830.245 a	338.007	176.041	<1%	0.955
	C	32,587.113 e	931.060	19.188	<1%	0.700
	h°	100,934.704 f	2883.849	64.704	<1%	0.887
	TSS	229.323 g	6.552	18.106	<1%	0.688
	pH	15.453 h	0.442	32.663	<1%	0.799
	TA	8.523 i	0.244	160.312	<1%	0.951
	TPC	1,688,292.788 j	48,236.937	62.301	<1%	0.883
	TAC	314,898,910 k	8997.112	13.864	<1%	0.628
	TSC	3,775,162 l	107.862	5.851	<1%	0.416
Intercept	Weight loss	24,142.989	24,142.989	12,574.184	<1%	0.978
	C	480,995.758	480,995.758	9912.810	<1%	0.972
	h°	280,525.889	280,525.889	6294.068	<1%	0.956
	TSS	59,685.204	59,685.204	164,929.691	<1%	0.998
	pH	4831.045	4831.045	357,395.29	<1%	0.999
	TA	32.881	32.881	21,645.720	<1%	0.987
	TPC	12,885,166.586	12,885,166.586	16,642.050	<1%	0.983
	TAC	1,342,058.994	1,342,058.994	2067.984	<1%	0.878
	TSC	2,126,768.706	2,126,768.706	115,372.72	<1%	0.998
Cultivar	Weight loss	135.882	135.882	70.770	<1%	0.197
	C	312.068	312.068	6.431	<5%	0.022
	h°	89,708.869	89,708.869	2012.769	<1%	0.875
	TSS	7.871	7.871	21.751	<1%	0.070
	pH	9.656	9.656	714.330	<1%	0.713
	TA	7.868	7.868	5179.816	<1%	0.947
	TPC	510,348.809	510,348.809	659.149	<1%	0.696
	TAC	12,809.947	12,809.947	19.739	<1%	0.064
	TSC	5.700	5.700	0.309	>5%	0.001
Treatment	Weight loss	1266.893	1266.893	659.825	<1%	0.696
	C	4934.138	4934.138	101.687	<1%	0.261
	h°	215.859	215.859	4.843	<1%	0.017
	TSS	68.062	68.062	188.079	<1%	0.395
	pH	0.383	0.383	28.320	<1%	0.090
	TA	0.068	0.068	44.559	<1%	0.134
	TPC	1192.744	1192.744	1.541	>5%	0.005
	TAC	75,822.526	75,822.526	116.835	<1%	0.289
	TSC	81.939	81.939	4.445	<5%	0.015
Storage duration	Weight loss	9435.039	1179.380	614.246	<1%	0.945
	C	22,727.513	2840.939	58.549	<1%	0.619
	h°	5614.685	701.836	15.747	<1%	0.304
	TSS	16.594	2.074	5.732	<1%	0.137
	pH	2.742	0.343	25.361	<1%	0.413
	TA	0.381	0.048	31.380	<1%	0.466
	TPC	1,051,375.562	131,421.945	169.740	<1%	0.825
	TAC	141,285.696	17,660.712	27.213	<1%	0.431
	TSC	1837.576	229.697	12.461	<1%	0.257
Cultivar * Treatment	Weight loss	128.333	128.333	66.838	<1%	0.188
	C	1510.982	1510.982	31.140	<1%	0.098
	h°	1547.603	1547.603	34.723	<1%	0.108
	TSS	37.007	37.007	102.262	<1%	0.262
	pH	0.117	0.117	8.644	<1%	0.029
	TA	0.060	0.060	39.390	<1%	0.120
	TPC	9742.451	9742.451	12.583	<1%	0.042
	TAC	31,257.721	31,257.721	48.165	<1%	0.143
	TSC	271.207	271.207	14.712	<1%	0.049
Cultivar * Storage duration	Weight loss	325.842	40.730	21.213	<1%	0.371
	C	574.900	71.863	1.481	>5%	0.040
	h°	1973.727	246.716	5.535	<1%	0.133
	TSS	74.483	9.310	25.728	<1%	0.417
	pH	2.227	0.278	20.590	<1%	0.364
	TA	0.100	0.013	8.242	<1%	0.186
	TPC	48,444.104	6055.513	7.821	<1%	0.178
	TAC	20,432.389	2554.049	3.936	<1%	0.099
	TSC	1036.081	129.510	7.026	<1%	0.163
Treatment * Storage duration	Weight loss	421.075	52.634	27.413	<1%	0.432
	C	1600.473	200.059	4.123	<1%	0.103
	h°	681.276	85.159	1.911	>5%	0.050
	TSS	15.653	1.957	5.407	<1%	0.131
	pH	0.199	0.025	1.839	>5%	0.049
	TA	0.022	0.003	1.819	>5%	0.048
	TPC	33,110.319	4138.790	5.346	<1%	0.129
	TAC	21,803.731	2725.466	4.200	<1%	0.104
	TSC	348.024	43.503	2.360	<1%	0.062
Cultivar * Treatment * Storage duration	Weight loss	117.181	14.648	7.629	<1%	0.175
	C	927.039	115.880	2.388	<5%	0.062
	h°	1192.685	149.086	3.345	<1%	0.085
	TSS	9.653	1.207	3.334	<1%	0.085
	pH	0.130	0.016	1.198	>5%	0.032
	TA	0.024	0.003	1.952	>5%	0.051
	TPC	34,078.800	4259.850	5.502	<1%	0.133
	TAC	11,486.900	1435.863	2.213	<5%	0.058
	TSC	194.635	24.329	1.320	>5%	0.035

*: Interaction between factors. a. R Squared = 0.955 (Adjusted R Squared = 0.950). e. R Squared = 0.700 (Adjusted R Squared = 0.663). f. R Squared = 0.887 (Adjusted R Squared = 0.873). g. R Squared = 0.688 (Adjusted R Squared = 0.650). h. R Squared = 0.799 (Adjusted R Squared = 0.774). i. R Squared = 0.951 (Adjusted R Squared = 0.945). j. R Squared = 0.883 (Adjusted R Squared = 0.869). k. R Squared = 0.628 (Adjusted R Squared = 0.582). l. R Squared = 0.416 (Adjusted R Squared = 0.345).

**Table 2 foods-14-00337-t002:** Mean values ± SE for color attributes (C* (chroma value, color saturation) and h° (hue value, color angle)) during the storage time of ‘Pomegranate’ cultivars.

Cultivar	Storage Duration	Chroma		Hue°	
		Control	IMZ	Control	IMZ
Kingdom	T0	55.27 ± 3.93	55.27 ± 3.93	21.52 ± 4.5	21.52 ± 4.51
	T1	55.27 ± 3.93	48.54 ± 2.39	21.51 ± 4.51	26.96 ± 2.67
	T2	55.27 ± 3.93	35.87 ± 9.89	21.52 ± 4.51	11.17 ± 7.7
	T4	44.7 ± 6.21	33.99 ± 15.24	14.714 ± 5.31	8.8 ± 3.35
	T4	44.7 ± 6.21	29.82 ± 9.39	14.71 ± 5.3	6.9 ± 8.42
	T5	42.73 ± 6.36	31.09 ± 1.82	15.36 ± 6.21	7.5 ± 1.1
	T6	41.83 ± 6.68	22.32 ± 8.66	9.71 ± 6.1	3.01 ± 7.31
	T7	37.15 ± 11.17	23.89 ± 0.55	11.92 ± 8.8	0.91 ± 0.48
	T8	33.22 ± 6.52	20.25 ± 4.08	11.13 ± 6.99	1.29 ± 1.21
	Signific. (*p* ≤ F)	<0.001	<0.001	<0.001	<0.001
Sefri Ouled Abdellah	T0	47.43 ± 7.1	47.43 ± 7.1	47.31 ± 9.75	47.31 ± 9.75
	T1	46.96 ± 4.23	51.29 ± 1.84	47.55 ± 6.03	50.26 ± 4.71
	T2	44.99 ± 6.22	47.9 ± 5.3	45.4 ± 6.5	48.95 ± 7.79
	T4	39.82 ± 8.53	34.63 ± 2.69	49.29 ± 6.22	47.55 ± 2.36
	T4	39.22 ± 3.5	35.16 ± 6.75	48.89 ± 3.9	45.86 ± 2.59
	T5	39.02 ± 2.81	25.38 ± 8.51	45.29 ± 3.96	47.11 ± 3.63
	T6	35.88 ± 11.95	26.63 ± 7.24	43.69 ± 11.27	47.56 ± 6.2
	T7	32.57 ± 6.53	26.82 ± 12.6	42.51 ± 8.21	46.43 ± 14.69
	T8	27.73 ± 5.45	27.01 ± 1.75	32.27 ± 12.76	45.88 ± 1.81
	Signific. (*p* ≤ F)	<0.001	<0.001	<0.001	<0.001

## Data Availability

The raw data supporting the conclusions of this article will be made available by the authors upon request.

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
