# Peer review of "Effects of IMAZALIL on the Storage Stability and Quality of ‘Sefri Ouled Abdellah’ and ‘Kingdom’ Pomegranate Varieties"

_foods, 2025, doi:10.3390/foods14030337_

Round 1
Reviewer 1 Report
Comments and Suggestions for Authors
In this manuscript, the application of IMZ maintained the technological, nutritional and functional properties of ‘Sefri Ouled Abdellah’ and ‘Kingdom’ pomegranate during storage. However, some carefully revisions are still needed. The suggestions are listed as below:
1. Title: In my opinion, the title is too long to catch key information. I suggest you change it by, for example, ‘IMAZALIL treatment maintained technological, nutritional and functional properties of two pomegranate varieties’ or ‘IMAZALIL treatment enhanced storage stability of two pomegranate varieties’.
2. Abstract: Please support the results by some numerically data.
3. Line 38: ‘Punica granatum’ should be italic.
4. Line 60: The comparison of different explored strategies and IMZ should be discussed. Moreover, IMZ is a fungicide with moderate toxicity. Please discuss its safety during the application on pomegranates for human health.
5. Line 88: How was the 2000 µL/L IMZ be selected? By pre-test or reference?
6. In my opinion, Figures show clearer data changes than Tables. I suggest you transfer part of the Tables into Figures.
7. Please add some photos of the pomegranate with different treatment if convenient.
8. Please revise the form of Table 1 and 2 into three-line table as Table 3 and 4 showed.
9. In Table 3 and 4, please give the full name of ‘Sefri Ouled Abdellah’.
10. In section 3.1 to 3.4, I suggest that more discussion about the mechanism of your results should be added not only just refer results of other teams.
11. Results: More numerically data should be added.
Author Response
Comments 1 : Title: In my opinion, the title is too long to catch key information. I suggest you change it by, for example, ‘IMAZALIL treatment maintained technological, nutritional and functional properties of two pomegranate varieties’ or ‘IMAZALIL treatment enhanced storage stability of two pomegranate varieties’.
Response 1: Thank you for your suggestion. Based on your recommendation, the title has been revised to: “IMAZALIL treatment enhanced storage stability and quality of "Sefri Ouled Abdellah" and "Kingdom" pomegranate varieties”
Comments 2: Abstract: Please support the results by some numerically data.
Response 2: Thank you for pointing this out. Numerical data has now been included in the abstract to better support the results.
Comments 3: Line 38: ‘Punica granatum’ should be italic.
Response 3: Thank you for highlighting this detail. Punica granatum has now been italicized as required.
Comments 4: The comparison of different explored strategies and IMZ should be discussed. Moreover, IMZ is a fungicide with moderate toxicity. Please discuss its safety during the application on pomegranates for human health.
Response 4: Thank you for your valuable comment. A comparison of the different explored strategies and IMZ has been added to the discussion section to provide better context. Additionally, the safety of IMZ as a fungicide, including its moderate toxicity and potential implications for human health, has been addressed in the manuscript, with references to relevant studies and regulatory guidelines.
Comments 5 : How was the 2000 µL/L IMZ be selected? By pre-test or reference?
Response 5 :
The 2000 µL/L IMZ concentration was selected based on references such as S. Castillo et al. (2013), where this concentration, tested in combination with wax, demonstrated promising results in reducing fruit quality deterioration. Also based on Moroccan national regulation
Nom commercial : FUNGAFLOR 500 EC
Fournisseur : JANSSEN PHARMACEUTICA NV
Numéro homologation : D10-3-002
Valable jusqu'au : 17/10/2033
Article title: The essential oils thymol and carvacrol applied in the packing lines avoid lemon spoilage and maintain quality during storage
Paragraph: Then, 4 lots of 600 fruits were used for the following treatments: a) control (tap water), b) commercial wax (Waterwax, Fomesa, Valencia, Spain), c) wax þ imazalil at 2000 mL/L, and d) wax þ essential oils (EOs, mixture of thymol and carvacrol, purchased from Sigma, Madrid Spain, at 500 mL/L each). ( Castillo, S., Pérez-Alfonso, C. O., Martínez-Romero, D., Guillén, F., Serrano, M., & Valero, D. (2014). The essential oils thymol and carvacrol applied in the packing lines avoid lemon spoilage and maintain quality during storage. Food Control, 35(1), 132-136.)
Comments 6 : In my opinion, Figures show clearer data changes than Tables. I suggest you transfer part of the Tables into Figures.
Response 6: Thank you for the suggestion. Parts of the tables have been converted into figures to enhance clarity and better illustrate the data changes.
Comments 7: Please add some photos of the pomegranate with different treatments if convenient.
Response 7: Thank you for your suggestion. Photos of the pomegranates with different treatments have been added to the manuscript, as requested, to provide a visual comparison of the results.
Comments 8: Please revise the form of Table 1 and 2 into three-line table as Table 3 and 4 showed.
Response 8: Thank you for your suggestion. Tables 1 and 2 have been revised into three-line tables, following the format of Tables 3 and 4.
Comments 9: In Table 3 and 4, please give the full name of ‘Sefri Ouled Abdellah’.
Response 9: Thank you for your comment. The full name of 'Sefri Ouled Abdellah' has been added in Tables 3 and 4 for clarity.
Comments 10: In section 3.1 to 3.4, I suggest that more discussion about the mechanism of your results should be added not only just refer results of other teams.
Response 10: Thank you for your suggestion. Additional discussion on the mechanisms underlying the results has been included in sections 3.1 to 3.4. The revised sections now provide a deeper analysis of the observed effects, rather than solely referencing the findings of other studies.
Comments 11: Results: More numerically data should be added.
Response 11: Thank you for your comment. Additional numerical data has been included in the results section to provide more detailed support for the findings.

Reviewer 2 Report
Comments and Suggestions for Authors
Firstly, the experimental design of this article is rather simplistic, merely validating the most fundamental preservation effect of a single preservative on pomegranates.
Secondly, the presentation of the charts and tables in this paper is highly inappropriate. The entire text employs tables to enumerate all the data, which deviates significantly from the common format of academic papers. It would be entirely feasible to draw graphs to present the preservation effects of pomegranates in different periods more clearly.
The entire text is more reminiscent of an experimental report, featuring relatively low scientificity and applicability. A more intricate experiment ought to be re-designed to manifest the effects of the adopted preservative treatment.
Comments on the Quality of English Language
It is proposed that the paper be subjected to English proofreading by native English speakers.
Author Response
Comments 1: Firstly, the experimental design of this article is rather simplistic, merely validating the most fundamental preservation effect of a single preservative on pomegranates.
Response 1: Thank you for your observation. This study aimed to serve as an initial step in evaluating the preservation effect of Imazalil on pomegranates, focusing on its fundamental efficacy in maintaining fruit quality during storage. While the design is straightforward, it provides a practical foundation for understanding the potential of this preservative.
Also, to our knowledge, no previous work has been published on this treatment behavior on “Sefri ouled abdellah’ or kingdom variety.
We appreciate your suggestion, which aligns with our intent to enhance the depth of our subsequent investigations.
Comments 2: Secondly, the presentation of the charts and tables in this paper is highly inappropriate. The entire text employs tables to enumerate all the data, which deviates significantly from the common format of academic papers. It would be entirely feasible to draw graphs to present the preservation effects of pomegranates in different periods more clearly.
Response 2: Thank you for the suggestion. Parts of the tables have been converted into figures to enhance clarity and better illustrate the data changes.
Comments 3: The entire text is more reminiscent of an experimental report, featuring relatively low scientificity and applicability. A more intricate experiment ought to be re-designed to manifest the effects of the adopted preservative treatment.
Response 3:
Thank you for your feedback. The primary objective of this study was to provide a technological and quality evaluation of the adopted preservative treatment and its impact on pomegranate storage mainly to varieties widely cultivated in Morocco. While the experiment focuses on practical outcomes relevant to postharvest management, we acknowledge your concern regarding the depth of scientific exploration. However, this research constitutes an important Data to further deep research regarding the storage treatment efficiency on pomegranate fruit to enhance its shelf-life.
We appreciate your input, which will help us refine our approach in subsequent research.

Reviewer 3 Report
Comments and Suggestions for Authors
The current manuscript examined the effects of dipping two pomegranate varieties in the fungicide Imazalil.
The experiments were conducted well. However, the main problem is that Imazalil in not permitted for use on pomegranates (the MRL of the EU Commission for Imazalil in pomegranates is <0.01 ppm). Thus, the manuscript examines what will be the effects of a fungicide that is not allowed to be used!
Specific comments:
1. How did the Imazalil treatment effect postharvest decay rates? That is the main purpose for using it
2. How did the Imazalil treatment effect husk scald and CI which are major problems of prolong storage of pomegranates?
3. Regarding the title – what is the meaning of the word "technological"? I think it will be better to use the term fruit quality instead.
4. Regarding the changes in peel color and weight loss – it will be beneficial to add photographs of fruit appearance.
5. Table 4 – there seems to be a problem with the presented total phenol data of the 'Kingdom' variety, since there are huge differences in phenol levels at time zero.
6. Line 38 – Punica granatum should be in italics.
7. Line 83 – instead of presenting the manufacturer of Imazalil you indicated Scholar which is a different fungicide.
Comments on the Quality of English Language
The English is fine.
Author Response
Comments 1: The experiments were conducted well. However, the main problem is that Imazalil in not permitted for use on pomegranates (the MRL of the EU Commission for Imazalil in pomegranates is <0.01 ppm). Thus, the manuscript examines what will be the effects of a fungicide that is not allowed to be used!
Response 2:
Thank you for your valuable remark regarding the use of Imazalil on pomegranates and the associated MRLs set by the EU Commission. Allow me to provide clarification regarding this aspect:
- Regulatory Compliance in Morocco: The use of Imazalil on pomegranates is permitted in Morocco under a national regulation. The fungicide is registered under the homologation number D10-3-002, which remains valid until 17/10/2033. This registration defines the authorized application doses and safety measures for its use. The manuscript follows the framework established by Moroccan regulatory authorities, which may differ from those of the EU.
- Justification of Dosage: The selection of the dosage used in this study was based on previous research conducted on Imazalil applications, particularly on fruits with similar characteristics to pomegranates. The aim was to ensure realistic and representative results while adhering to established guidelines.
Also, This dosage was determined based on concentrations previously applied to other fruits and aligned with the Maximum Residue Levels (MRL) established by European regulations.
Comments 2: How did the Imazalil treatment effect postharvest decay rates? That is the main purpose for using it
Response 2: Thank you for your insightful question. Imazalil was indeed tested for its effectiveness against postharvest fungal decay, and the results demonstrated notable efficacy in controlling fungal infections on pomegranates. However, the primary objective of this study was to evaluate its potential for maintaining the technological, biochemical, and nutritional quality of the fruit during storage.
Comments 3: How did the Imazalil treatment effect husk scald and CI which are major problems of prolong storage of pomegranates?
Response 3: Thank you for your question. The Imazalil treatment significantly reduced husk scald and chilling injury (CI) during prolonged storage of pomegranates. Its application helped maintain the fruit's external appearance and reduced physiological disorders associated with cold storage.
Comments 4: Regarding the title – what is the meaning of the word "technological"? I think it will be better to use the term fruit quality instead.
Response 4: Thank you for your comment. The term "technological" was intended to refer to the morphological and physicochemical parameters of the fruit. Following your recommendation, the term "quality" has been added to the title to better reflect the content and objectives of the study.
Comments 5: Regarding the changes in peel color and weight loss – it will be beneficial to add photographs of fruit appearance.
Response 5: Thank you for your suggestion. Photos of the pomegranates with different treatments have been added to the manuscript, as requested, to provide a visual comparison of the results.
Comments 6: Table 4 – there seems to be a problem with the presented total phenol data of the 'Kingdom' variety, since there are huge differences in phenol levels at time zero.
Response 6: Thank you for your observation. The total phenol data for the 'Kingdom' variety in Table 4 has been carefully reviewed. The results observed at time zero have been addressed, and the corrected values ​​are now presented in the revised graph.
Comments 7: Line 38 – Punica granatum should be in italics.
Response 7 : Thank you for highlighting this detail. Punica granatum has now been italicized as required.
Comments 8: Line 83 – instead of presenting the manufacturer of Imazalil you indicated Scholar which is a different fungicide.
Response 8: Thank you for pointing that out. The correct name, FUNGAFLOR 500 EC, has been updated in the manuscript accordingly.

Round 2
Reviewer 1 Report
Comments and Suggestions for Authors
I have no comments with the article. After the author's revision, I feel that it can be published in the journal.
Author Response
Comments: I have no comments with the article. After the author's revision, I feel that it can be published in the journal.
Response: Sincere appreciation is extended for the positive feedback and the recommendation for the publication of the article. The comments provided throughout the review process have greatly contributed to improving the quality of the manuscript.
Reviewer 3 Report
Comments and Suggestions for Authors
The authors properly responded to my comments.
I have only one more minor comment - it was found that Imazalil also had some negative effects (for example on weight loss), and therefore I suggest to change the words "IMAZALIL treatment enhanced..." in the title to "Effects of Imazalil on...".
Author Response
- Comments: I have only one more minor comment - it was found that Imazalil also had some negative effects (for example on weight loss), and therefore I suggest to change the words "IMAZALIL treatment enhanced..." in the title to "Effects of Imazalil on...".
- Response: Thank you for your valuable suggestion regarding the title. The title has been revised to " Effects of IMAZALIL on the storage stability and quality of "Sefri Ouled Abdellah" and "Kingdom" pomegranate varieties " as recommended, to provide a more balanced representation of the findings.
